# Cerebrovascular risk factors impact frontoparietal network integrity and executive function in healthy ageing

Michele Veldsman [1,2,6 ✉], Xin-You Tai[3,4,6], Thomas Nichols [5], Steve Smith [1,3], João Peixoto[2], Sanjay Manohar [2,3,4] & Masud Husain [1,2,3,4]

Healthy cognitive ageing is a societal and public health priority. Cerebrovascular risk factors increase the likelihood of dementia in older people but their impact on cognitive ageing in younger, healthy brains is less clear. The UK Biobank provides cognition and brain imaging measures in the largest population cohort studied to date. Here we show that cognitive abilities of healthy individuals (N = 22,059) in this sample are detrimentally affected by cerebrovascular risk factors. Structural equation modelling revealed that cerebrovascular risk is associated with reduced cerebral grey matter and white matter integrity within a fronto-parietal brain network underlying executive function. Notably, higher systolic blood pressure was associated with worse executive cognitive function in mid-life (44–69 years), but not in late-life (>70 years). During mid-life this association did not occur in the systolic range of 110–140 mmHg. These findings suggest cerebrovascular risk factors impact on brain structure and cognitive function in healthy people.

[1] Wellcome Centre for Integrative Neuroimaging, University of Oxford, Oxford, UK. [2] Department of Experimental Psychology, University of Oxford, Oxford, UK. [3] Nuffield Department of Clinical Neuroscience, University of Oxford, Oxford, UK. [4] Division of Clinical Neurology, John Radcliffe Hospital, Oxford University Hospitals Trust, Oxford, UK. [5] Nuffield Department of Population Health, University of Oxford, Oxford, UK. [6]These authors contributed equally: Michele Veldsman, Xin-You Tai. ✉email: michele.veldsman@psy.ox.ac.uk

As the world's ageing population becomes one of the most significant societal transformations of the twenty-first century[1], understanding healthy cognitive ageing has become a critical public health priority. Two key questions have emerged. First, are there modifiable factors that can protect or improve cognition as people age? Second, what changes within brain networks underpin cognitive decline as humans grow older[2]? Several lines of evidence suggest that cerebrovascular risk (CVR) factors might potentially have a significant impact on cognitive trajectories[3] and are associated with neuroimaging markers that are considered to have disruptive effects on brain networks critical for cognitive function[4]. However, the strength of inferences that can be made on the basis of current findings is quite limited, either because of relatively small sample sizes or highly variable results. Moreover, many studies have used dementia or pre-dementia (mild cognitive impairment, MCI) as primary outcomes[5], and are based on relatively insensitive clinical measures such as the mini-mental state examination (MMSE). Such indices would not be capable of detecting some of the mild cognitive changes that occur during ageing, which can be subtle but nevertheless significantly impact daily quality of life[6].

Notwithstanding these limitations, the results from several different cohorts suggest that CVR factors might be an important target to protect brain health and thereby reduce the risk of developing dementia. For example, a community-based study followed 6330 non-demented individuals, aged 45–99 years, and showed that mid-life diabetes and impaired glucose tolerance conferred increased risk of dementia[7,8]. More recently, a community-based investigation in the United States reported that mid-life hypertension is a significant risk factor for later-life dementia ($N = 4761$)[9]. However, a study of 465 UK individuals, aged 69–71 years born on the same week, found no evidence that blood pressure (BP) during early adulthood into mid-life affected pre-clinical Alzheimer cognitive scores[10] or cerebral amyloid-β pathology, measured by positron emission tomography, despite showing an association with reduced brain volumes. Therefore, the results of several investigations of hypertension and risk of developing dementia have been inconsistent[11]. Other risk factors variably implicated in increasing dementia risk in the literature include body-mass index, hypertriglyceridemia and smoking[12,13].

In addition to these disparate findings relating to long-term dementia risk, the relationship between CVR factors and current cognitive function in people without dementia or MCI also remains unclear. Several studies have suggested an association between cerebrovascular burden and cognition in healthy cohorts (reviewed, for example here[14]). However, most investigations have assessed the presence of vascular risk factors or risk factor modification and related these to global cognition, using either MMSE or composite cognitive scores[15–19]. One recent study has reported a benefit of statin and BP-lowering medication using a more sensitive digit-symbol substitution score in healthy people without any known cardiovascular disease[20]. Overall, however, to the best of our knowledge there has been no clear demonstration of a parametric effect of risk factors (e.g., BP level, rather than simply presence or absence of hypertension or BP-lowering treatment) associated with systematic differences in cognitive performance. This represents an important gap in current knowledge.

In addition, the strength of any conclusion regarding the impact of CVR factors—or their modification—on cognition would be most convincing if they were shown to be accompanied by changes in brain structure, ideally in a large sample in which cognitive function was concurrently measured with tests that are more sensitive than the MMSE or other global measures of cognition. However, to date, we lack data from a large-scale investigation, which combines sensitive measure of cognition

with modern brain imaging markers to provide more definitive conclusions.

It is, of course, well established that CVR factors are associated closely with the macroscale marker white matter hyperintensities (WMH)[21] on magnetic resonance imaging (MRI). WMH have generated significant interest as a key marker of cerebrovascular burden in the ageing brain[22,23]. However, reports of their effect on cognitive function have been variable. While some studies have concluded that there is a deleterious association between WMH and speed of processing or working memory[24,25], others showed no such relationship[26]. Advances in neuroimaging now allow more precise estimation of microscale degeneration of grey and white matter in complex brain networks. The frontoparietal network is especially vulnerable to grey matter atrophy[27] and white matter degeneration[28] in aging[29,30], associated with significant cognitive consequences[28,31]. White matter degeneration has specific cognitive effects, with the integrity of tracts connecting frontal and parietal regions impacting upon executive cognitive function, while integrity of tracts connecting posterior and visual regions is related to visual memory[28]. Integration of advanced MRI measures of microstructural network integrity therefore has the potential to provide early and sensitive markers of the impact of cerebrovascular burden, and may improve mechanistic models of its neurocognitive consequences. However, few studies have leveraged the combination of clinical assessment, task-based cognitive tests and multi-modal neuroimaging in large populations.

To summarise, the existing literature leaves several important unanswered questions. What are the effects of CVR factors on cognitive function in healthy people across the lifespan? And is it possible to show a relationship, in a parametric manner, between a risk factor such as BP level and cognition? Does treatment of a modifiable risk factor, such as hypertension, affect cognitive function? If so, is there a range of BP over which this treatment has an impact? Finally, does the integrity of a distributed brain network underlying a specific cognitive function explain decline in that function across the lifespan as a result of cerebrovascular burden?

Here, in the most extensive cross-sectional study to date, we investigated data from the UK Biobank, a large population cohort of over 500,000 middle-to-older age individuals who underwent medical, sociodemographic and cognitive assessment between 2006 and 2010[32]. We examined data from 22,059 individuals from the UK Biobank who also attended for multi-modal neuroimaging. Previous studies have examined the relationship between either cardiovascular risk and brain health, or between brain health and cognition, or between cognition and individual cardiovascular risk factors. Given our large sample, we were able to include all these variables into one large, complex model that allowed us simultaneously to examine multiple relationships and establish how CVR and grey and white matter integrity in a frontoparietal network relates to cognitive decline, specifically with respect executive function, in healthy ageing.

Our study objectives were threefold. First, we aimed to better understand the relationship between CVR factors and cognition, by characterising the effect of each risk factor on a continuous measure of executive function. Second, we investigated the potential effects of risk factor modification, focussing on how hypertension treatment affects the relationship between systolic blood pressure (SBP) and executive function. Third, we tested the role of the frontoparietal network in the relationship between age, CVR factors, WMH and executive function using structural equation modelling (SEM). Semi-partial correlation and mediation analysis were used to quantify directional relationships between MRI markers of brain health and their impact on cognition to develop a mechanistic model of neurocognitive aging.

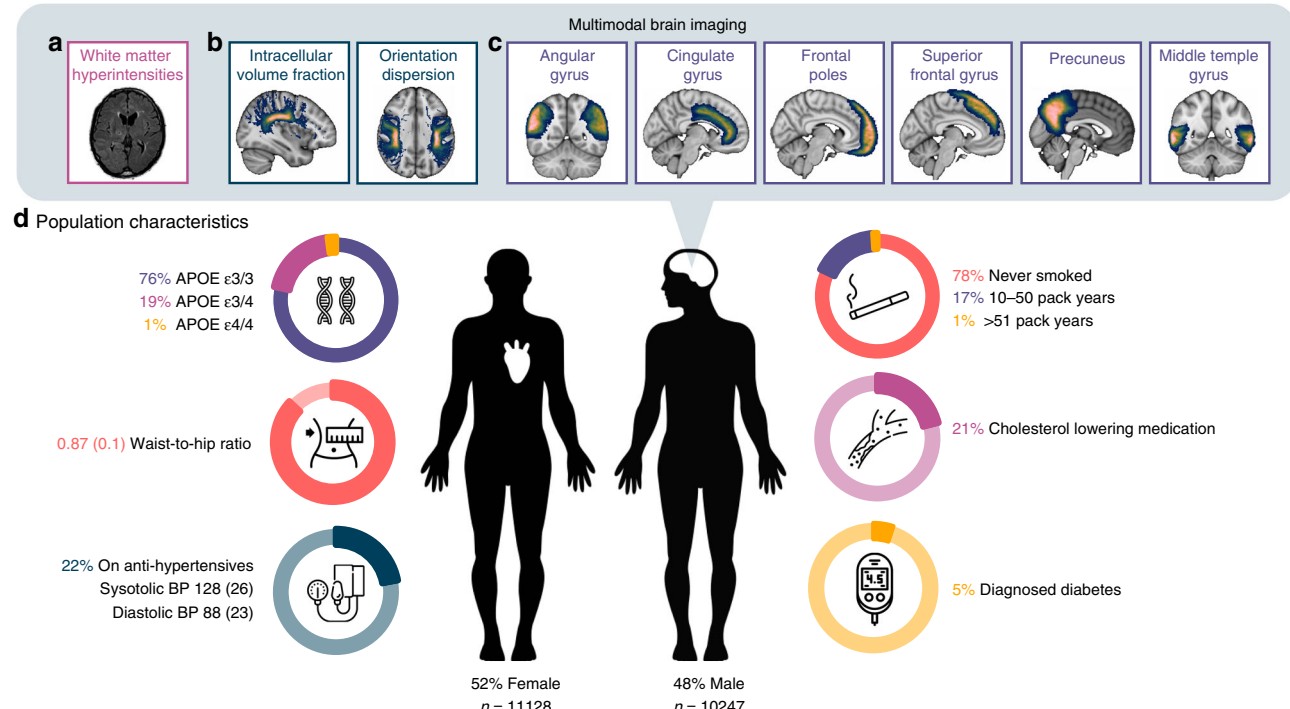

**Fig. 1 Population characteristics and multi-modal imaging of 22,059 healthy ageing adults.** Multi-modal imaging—**a** estimation of white matter hyperintensity load, **b** measures of white matter integrity and **c** grey matter volume in the frontoparietal network. **d** Population characteristics—including mean systolic and diastolic blood pressure in mmHg (standard deviation). These characteristics are used to derive a total cerebrovascular burden risk score. APOE apolipoprotein E; BP blood pressure.

## Results

**Population characteristics of 22,059 healthy ageing adults.** Clinical, demographic, cognitive and brain imaging data were analysed in 22,059 people from the UK Biobank aged 44–73 (mean age 62, SD 7) at the time of their MRI scan (Fig. 1, Supplementary Table 1). The apolipoprotein-E (*APOE*) allele frequency, an established genetic risk factor for Alzheimer's disease, within our cohort was 76% with ε3/3 allele, 19% with one ε4 allele and 1% having ε4/4 genotype, similar to expected numbers of a representative Caucasian population[33]. Five percent of the cohort were diabetic, 21% were on cholesterol lowering medication and 22% were on antihypertensive medication. Smoking rates were relatively low, reflecting UK Biobank characteristics, with 78% classified as non-smokers[34].

**Cerebrovascular risk predictors of executive function.** We used confirmatory factor analysis (CFA) to estimate a continuous latent variable representing executive function. This latent variable predicted performance in the simple reaction time task and pairs matching task (standardised beta estimates 0.49 and 0.25, respectively, $p < 0.001$, see Supplementary Table 4). Multiple regression with the executive function latent variable as the dependent variable and with age, socio-economic status and individual CVR factors as covariates was significant, $F(8, 15,330) = 1010$, $p < 0.0001$, explaining 35% of the variance in executive function. Age was the strongest predictor (Table 1) followed by antihypertensive medication use, having diabetes and APOE status. We used these CVR factors to derive a risk score to represent total cerebrovascular burden within individuals and related it to measures derived from the multi-modal neuroimaging in our SEM.

**Relationship between blood pressure and executive function.** Further analysis focused on whether SBP, a continuous variable,

predicted executive function. In addition, we examined the effect of using antihypertensive medication, which can be considered to be a robust index of established hypertension. Sliding window analysis revealed a graded reduction in executive function performance with increasing SBP in individuals not taking antihypertensive medication (Fig. 2a). Participants on antihypertensives (with established hypertension) had lower corresponding executive function than those not on medication, significantly different when comparing correlation coefficients following Fisher's *r* to *z* transformation ($z = -3.87$, $p < 0.001$). For these individuals on antihypertensives, executive function was stable for the SBP range less than 140 mmHg, while SBP measurements >140 mmHg was associated with a decline in executive function.

We regressed out the confounding effects of age using quantile-based, age-residual analysis. Figure 2b shows the relationship between age and SBP while Fig. 2c displays the relationship between age and executive function. From this, we calculated and plotted the age residuals for SBP against executive function showing a significant relationship for those not taking antihypertensives (Pearson correlation $r = -0.155$, $p < 0.001$, Fig. 2d) and those who are on antihypertensives (Pearson correlation $r = -0.093$, $p < 0.001$, Fig. 2e), demonstrating significant relationships after controlling for age. For those on antihypertensives, however, the relationship was significantly weaker ($z = -3.84$, $p < 0.001$).

**Blood pressure and executive function in mid- and late-life.** Next, we examined the relationship between SBP and executive function, in the context of antihypertensive medication, in two age groups: mid-life (44–69 years) and late-life (>70 years) (Fig. 2f). The patterns observed above in the entire population (Fig. 2a) were present in the mid-life group, but crucially not in the late-life group. When controlling for age, the relationship

**Table 1 Cerebrovascular risk predictors of executive function.**

| Variable | Unstandardised beta estimate | Standard Error | Standardised beta estimate | t | p |
|---|---|---|---|---|---|
| Intercept | 2.86 | 0.06 | | 45.66 | 0.0001 |
| Waist-to-hip ratio | 0.03 | 0.06 | 0.003 | 0.50 | 0.616 |
| Smoking status | −0.03 | 0.01 | −0.02 | −2.92 | 0.004 |
| Medicated cholesterol | −0.03 | 0.01 | −0.02 | −2.42 | 0.016 |
| Medicated hypertension | −0.07 | 0.01 | −0.04 | −5.66 | 0.0001 |
| Diabetic | −0.09 | 0.02 | −0.03 | −4.05 | 0.0001 |
| APOE ε status | −0.04 | 0.01 | −0.02 | −3.60 | 0.0003 |
| Age at assessment | −0.06 | 0.001 | −0.57 | −81.99 | 0.0001 |
| Socio-economic status | −0.01 | 0.002 | −0.03 | −3.86 | 0.0001 |

Multiple regression of cerebrovascular variables on executive function latent variable. Socio-economic status indexed by the Townsend Deprivation Index.
APOE apolipoprotein E.

between SBP and executive function was significant for the mid-life group for both individuals on antihypertensives ($r = -0.06$, $p = 0.001$) and those who were not on such medication ($r = -0.13$, $p < 0.001$) (Fig. 2g, h), with a significant difference between the correlation coefficients ($z = -3.63$, $p < 0.001$). For the late-life group, once age was controlled, there was no significant relationship between SBP and executive function, either for those on ($p = 0.743$) or not on antihypertensives ($p = 0.110$; difference between correlation coefficients was also not significant, $p = 0.106$) (Fig. 2i, j).

**Frontoparietal network integrity predicts executive function**. SEM was used to quantify the multivariate, directional relationships between age, CVR burden, WMH load, frontoparietal grey matter volume, white matter integrity and executive function with path model relationships based on the current literature. The frontoparietal network was selected due to close association with executive and cognitive control mechanisms[35]. The same analysis was performed on a control, language network (Supplementary Fig. 3).

In our full path model (Fig. 3), all relationships were in the expected direction. All paths were significant, $p < 0.001$, except for the path between CVR and frontoparietal white matter integrity. Increasing age resulted in reduced frontoparietal grey matter volume and frontoparietal white matter integrity, and increased WMH load, reflecting age-related brain changes. CVR was a better predictor of WMH load ($\beta = 0.13$, $p < 0.001$) than grey matter volume ($\beta = -0.07$, $p < 0.001$) or white matter integrity ($\beta = -0.01$, $p = 0.26$, Supplementary Table 4). The path between CVR and frontoparietal white matter integrity would be significant if there were no path from WMH load to frontoparietal white matter integrity. In line with our hypotheses, frontoparietal grey and white matter integrity were strong predictors of executive function. The strength of the relationship between frontoparietal white matter integrity and executive function ($\beta = 0.34$, $p < 0.001$) and frontoparietal grey matter volume and executive function ($\beta = 0.32$, $p < 0.001$) were similar, suggesting important roles for both aspects of the frontoparietal network in maintaining executive function. Alongside robust model fit statistics, penalising overly complex models (Fig. 3b), the significant paths in the expected direction provide evidence that this model is a good fit for the data.

Model comparison indices provide evidence that the model, which includes both grey and white matter structural integrity of the frontoparietal network (Fig. 3b), better explains the relationship between cognition and WMH load than a model that excludes frontoparietal grey matter volume (Fig. 3 green panel, CFI = 0.73, RMSEA = 0.08) or excludes frontoparietal white matter integrity (Fig. 3 red panel, CFI = 0.83, RMSEA = 0.06). Formal comparison confirmed the full model was a better fit than either the

nested model constraining white matter integrity ($\chi^2(4) = 2745$, $p < 0.001$) or constraining grey matter volume ($\chi^2(4) = 4537$, $p < 0.001$). A model including a language network as a control was a poor fit (CFI = 0.82, TLI = 0.79, RMSEA = 0.06, Supplementary Table 5) for the data (Supplementary Fig. 3), confirming effects were specific to the frontoparietal network.

Semi-partial correlation and mediation were used to further interrogate the relationship between executive function and brain imaging variables. There was a significant negative correlation between WMH load and executive function ($r = -0.20$, $p < 0.001$; Fig. 3c). However, semi-partial correlation showed the magnitude of this relationship was significantly reduced when controlling for white matter integrity ($r = -0.07$, $p < 0.001$, Fig. 3d) and for grey matter volume ($r = -0.02$, $p < 0.01$, Fig. 3e). Therefore, much of the relationship between executive function and WMH load is actually explained by frontoparietal network integrity. This finding may help to resolve a longstanding debate as to the impact of WMH on executive function[36], suggesting at least part of this relationship is explained by microstructural changes and grey matter atrophy of a major network responsible for executive processing. To further clarify the relationship between frontoparietal grey matter volume and white matter integrity on executive function, mediation analysis was performed (Fig. 3f). We hypothesised that the relationship between grey matter volume and executive function was mediated by frontoparietal white matter integrity. In support of this proposal, partial mediation was found, with the relationship between executive function and frontoparietal grey volume reduced from a beta estimate 0.47–0.19 when controlling for white matter integrity (Fig. 3f).

**Discussion**
The results of this study show that combining CVR factors with measurements of cognitive function and neuroimaging markers of brain health leads to important and actionable insights into optimising cognitive health across normal ageing. Multivariate modelling allowed us to leverage extensive lifestyle, physiological, neuroimaging and genomic data to investigate the complex relationship between frontoparietal network integrity, WMH burden and executive function to facilitate a mechanistic understanding of how CVR factors affect cognitive health. Antihypertensive medication, diabetes and APOE status were all significant predictors of executive function. Effect sizes were significant despite being expectedly small.

In addition to identifying important associations between CVR and executive function, the potential mechanistic paths underpinning this process were examined statistically. SEM supported a pathway in which CVR factors mediate their impact on cognition through disruption of the distributed frontoparietal brain network underlying executive function (Fig. 3). This importantly shifts substantial focus from macroscale markers of CVR, namely

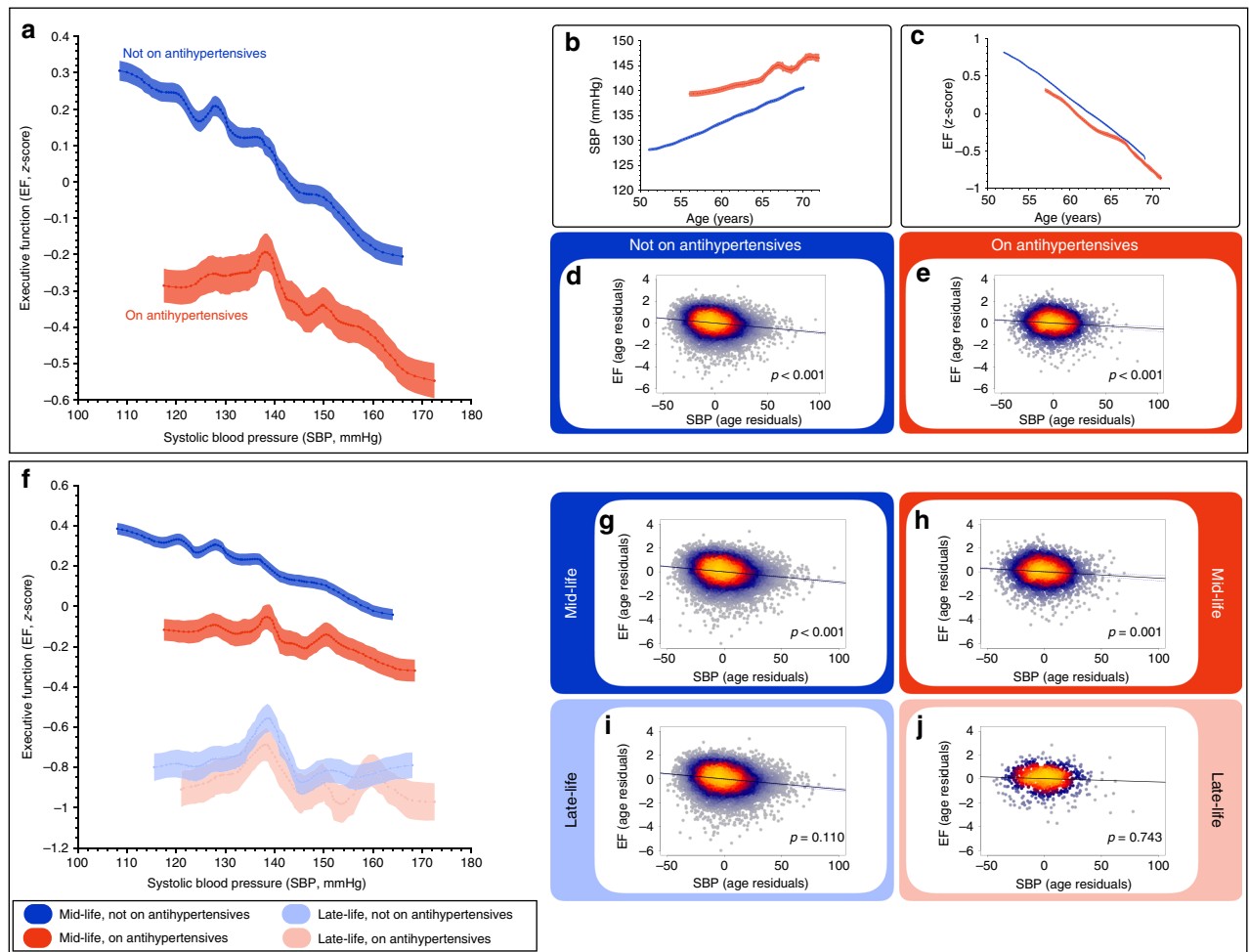

**Fig. 2 Relationship between blood pressure and executive function. a** Higher systolic blood pressure (SBP) was associated with lower executive function (EF) for both participants who were on antihypertensives and those who were not. For those on antihypertensives, there was a non-linear relationship as EF was not associated with a decline for SBP measurements <140 mmHg (EF standardised into z-score, **a–c**, **f**: shaded area represents standard error around the mean. **b**, **c** SBP increased with age, while EF declines with age, respectively, for participants who take antihypertensives and those who do not. **d**, **e** Association between EF and SBP using age residuals (adjusting for age) for participants who were not on antihypertensive medication (Pearson correlation, $r = -0.155$, 95% CI [$-0.172$, $-0.139$], $p < 0.001$, $n = 16,410$) and participants who are ($r = -0.093$, 95% CI [$-0.126$, $-0.060$], $p < 0.001$, $n = 4699$) with the difference between correlation coefficients being significant following Fisher's $r$ to $z$ transformation, $z = -3.84$, $p < 0.001$). **f** When considering age groups, higher SBP was associated with lower EF in mid-life (44–69 years) participants but not in the late-life (70–73 years) group (shaded area represents standard error). **g**, **h** Age-adjusted analysis between SBP and EF for participants not on an antihypertensives ($r = -0.134$, 95% CI [$-0.152$, $-0.115$], $p < 0.001$, $n = 13,242$) and those who were on antihypertensives ($r = -0.062$, 95% CI [$-0.100$, $-0.024$], $p = 0.001$, n = 3071), within the mid-life group (difference between group correlation coefficients, $z = -3.62$, $p < 0.001$). **l, j** Same relationship in the late-life group showing no significant correlation between SBP and executive function regardless of whether they were on antihypertensive medication ($r = -0.056$, 95% CI [$-0.1247$, $0.0129$], $p = 0.110$, $n = 1869$ and $r = 0.008$, 95% CI [$-0.0416$, $0.0582$], $p = 0.743$, $n = 949$). SBP systolic blood pressure; EF executive function; CI confidence interval.

WMHs, to more insidious degeneration of normal-appearing grey and white matter, which are already burdened through normal ageing[37]. A substantial literature has debated the association between executive function and WMH burden[36]. The data presented here statistically show that this relationship is at least in part explained by the microstructural integrity and volume of the frontoparietal network. The mediation analysis showed the relationship between frontoparietal grey matter volume and executive function was substantially mediated through frontoparietal white matter integrity. This result supports the hypothesised process of network-wide grey matter atrophy, likely reflecting secondary degeneration from disconnection due to white matter degeneration[38].

Risk factor modification of SBP had a significant effect on executive function. Overall, executive function declined with

increasing SBP (Fig. 2). Individuals who were on antihypertensives, i.e., with an established diagnosis of hypertension, showed an effective systolic range below ~140 mmHg corresponding to stable executive function while, above this threshold, increasing SBP was associated with a decline in executive function. Demonstrating this threshold and effect of SBP, in a parametric manner, on a continuous score of executive function was made possible by two unique factors of this study: the large sample size of the UK Biobank and the use of a latent variable based on neuropsychological testing, which is likely more sensitive than traditionally used total cognitive test scores. Intriguingly, this BP threshold actually reflects current American and European guidelines, which are based primarily on endpoints of cardiovascular events and associated morbidity[39,40].

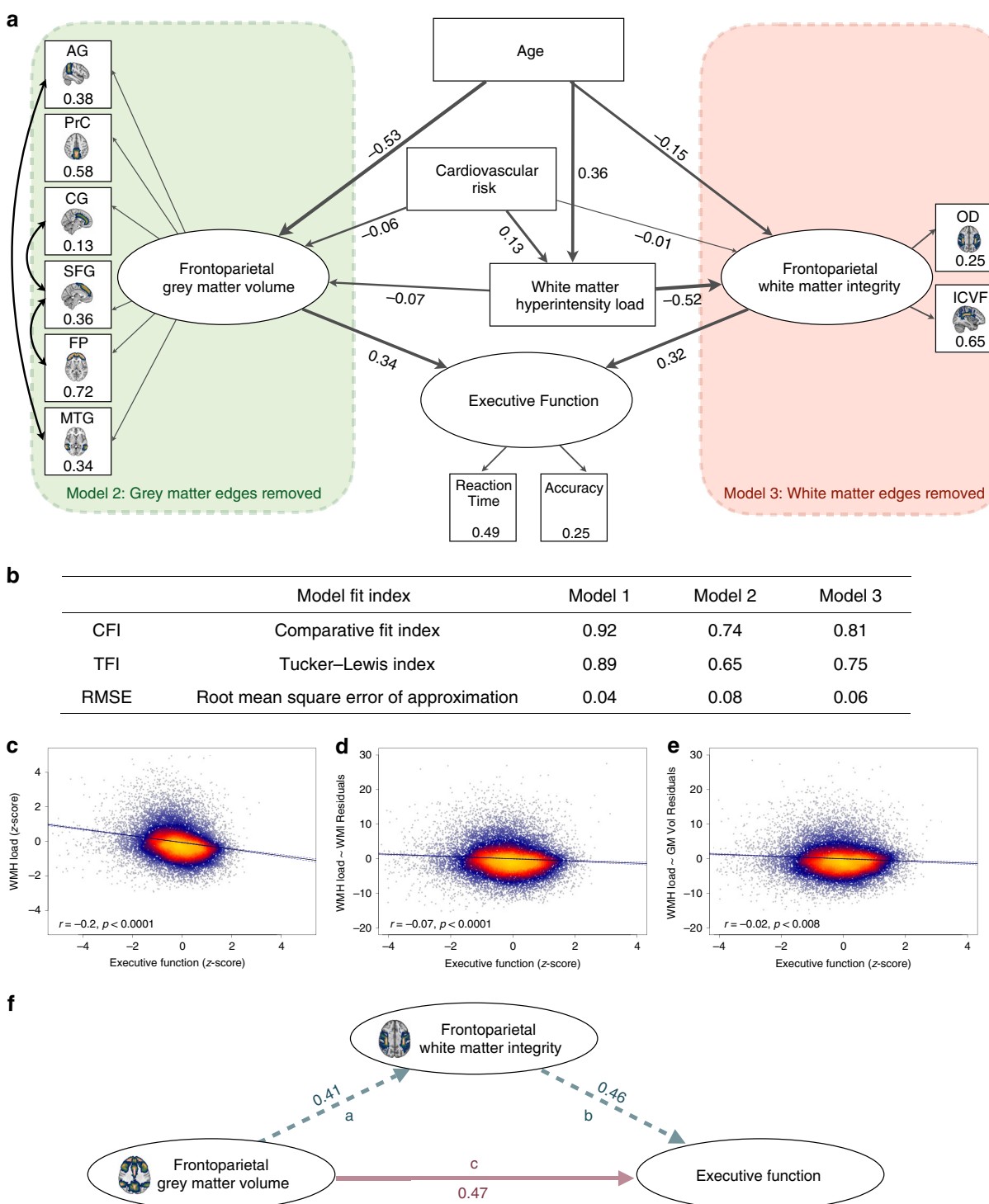

**Fig. 3 Structural equation model, semi-partial correlation and mediation analysis. a** Full structural model. All figures represent standardised beta coefficients. All paths significant to $p < 0.001$ except the path between cerebrovascular risk and frontoparietal white matter integrity. Black arrows represent covariances. Green box: indicates the frontoparietal grey matter edges that have been fixed to zero in Model 2 red box: the frontoparietal white matter integrity edges that have been fixed to zero in Model 3. **b** Model fit indices comparing nested models (green and red boxes). Semi-partial correlation plots showing relationship between **c** WMH load (z-score) and executive function (z-score) Pearson's $r = -0.20$, $p < 0.001$; **d** controlling for white matter integrity Pearson's $r = -0.07$, $p < 0.001$ and **e** controlling for grey matter volume Pearson's $r = -0.02$, $p < 0.008$. **f** Mediation analysis. Values are standardised beta estimates. Path a: relationship between frontoparietal white matter integrity and frontoparietal grey matter volume. Path b: relationship between executive function and frontoparietal white matter. Path c: direct relationship between frontoparietal grey matter volume and executive function. Figures in brackets show beta estimates for the indirect path in which the relationship between frontoparietal grey matter volume and executive function is mediated by frontoparietal white matter integrity. AG angular gyrus; PrC precuneus; CG cingulate gyrus; SFG superior frontal gyrus; FP frontal pole; MTG middle temporal gyrus; OD orientation dispersion; ICVF intracellular volume fraction; WMH white matter hyperintensity; GM Vol grey matter volume; WMI white matter integrity.

The findings of the current study have the potential to provide meaningful evidence on BP targets for cognitive health in normal ageing. They are in accord with recent findings, which emphasise the importance not simply to treat, but to treat effectively and monitor for response. The prospective SPRINT-MIND study[41] released results on the impact of intensive vs. standard BP control (defined as <120 mmHg vs. 120–140 mmHg, respectively) on cognitive impairment risk ($N = 9361$). This trial was terminated early and was considered underpowered. However, available results suggested no significant reduction in incidence of MCI[41] from aggressive BP control. The results presented in our investigation would be consistent. They show, in a parametric manner, that controlling SBP below 140 mmHg had no added benefit on executive function. Unlike previous studies which use BP levels to categorise groups, the findings presented here, using BP as a continuous variable, offer insights on a precise BP target for optimal cognitive health, albeit with the caveats associated with a cross-sectional study. They have potentially important implications for rational treatment guidance. SBP levels above 140 mmHg are associated with a measurable detrimental impact on current executive function. Conversely, systolic levels below 140 mmHg confer no additional benefit. Therefore, very aggressive treatment of BP risks hypotensive and other side effects, without necessarily providing a positive impact on cognition.

Further age-group analysis examining mid- and late-life groups showed striking differences. SBP predicted executive function within the mid-life group, but not in late-life individuals (aged > 70 years). While previous studies have suggested that higher systolic levels in mid-life may lead to worse cognitive performance in late-life[17,42], our results show the parametric benefit of hypertension treatment on current cognitive health in mid-life, but no significant effect of antihypertensives in late-life. This finding, in combination with the results of a recent study that examined dementia incidence associated with mid- to late-life BP patterns[9], emphasises the potential importance of treating mid-life hypertension for current cognitive function as well as for later dementia risk. Our data suggest a therapeutic age window to modify hypertension in relation to healthy cognitive ageing and provides a strong argument for robust BP assessments and clinical decision making for BP treatment in middle-age, otherwise healthy, individuals.

In the context of large prospective cohort analysis, the findings presented here should be assessed in relation to methodological and population considerations. Our observations were derived from cross-sectional data from the UK Biobank imaging cohort with fewer co-morbid diseases and from less deprived areas than the general population[34]. This selection bias may affect generalisability but also reflects a healthy ageing population with effects of CVR being potentially larger in more population-representative cohorts. The age range of UK Biobank participants may also lead to different environmental exposures during their lifespan. UK Biobank reliance on self-reported data may introduce information bias. However, we used surrogate markers, such as medication prescription to indicate established disease, to mitigate this when possible. Future release of prospective UK Biobank data will add a valuable temporal dimension to our interpretation, including longitudinal effect of antihypertensives on executive function and additional dementia outcomes.

In the rapidly growing social and health context of our ageing society, the findings presented here demonstrate the relationship between key CVR factors and healthy cognitive ageing, within the UK Biobank cohort, with implications on lifestyle practices for middle- and late-age populations. Moreover, the results are suggestive of targeted and time-sensitive BP control being beneficial to cognitive and brain health.

## Methods

**Participants.** The UK Biobank recruited 500,000 people aged 40–69 between 2006 and 2010, of which 100,000 will have brain imaging alongside the detailed health, demographic and cognitive assessments. MRI data from 22,059 people were available when the study was conducted under the UK Biobank application numbers 32,011 and 8107. All participants provided written, informed consent and the study was approved by the Research Ethics Committee (REC number 11/NW/0382).

We excluded individuals with a history or current diagnoses of neurological disease, brain injury, stroke, transient ischaemic attack, subdural or subarachnoid haematoma; infection of the nervous system; brain abscess, haemorrhage or skull fracture; encephalitis, meningitis, chronic neurological problem, amyotrophic lateral sclerosis, multiple sclerosis, Parkinson's or Alzheimer's disease, epilepsy, head injury, alcohol, opioid and other dependency according to the non-cancer illnesses codes (http://biobank.ndph.ox.ac.uk/showcase/coding.cgi?id=6).

**Cognitive testing.** Cognitive tests were administered via touchscreen on the same day as the MRI scan. Data from a pairs matching and simple reaction time task were analysed, indexing executive function and speed of processing, respectively[43]. In the pairs matching task, participants memorised the position of six pairs of matching cards presented simultaneously. The cards were then turned over and participants were required to touch the locations of the matching pairs with as few attempts as possible. The number of errors made was recorded. The simple reaction time task was a variation of the card game, Snap. Participants were required to press a button when two simultaneously presented cards matched. Mean reaction time over 12 rounds was recorded. Trials in which responses were below 50 ms were excluded as they reflected anticipation and trials over 2000 ms were excluded as the cards were no longer visible at this point.

**Cerebrovascular risk score.** We calculated a CVR score as the sum of known risk factors for cerebrovascular and brain health including hypertension, hypercholesteremia, smoking, diabetes, high waist-to-hip ratio (WHR) and *APOE* ε polymorphism status (Supplementary Fig. 1c). We did not use cardiovascular risk scores (such as the Framingham risk score[44]) as these require specific measures, such as high-density lipoprotein levels, which were not available from the UK Biobank at the time of analysis. These scores are used to calculate risk of future cardiovascular events, whereas we were interested in cerebrovascular burden on neurocognitive health.

BP was measured automatically on an Omron digital BP monitor, or a manual sphygmomanometer when the automated monitor was not available. The average across two readings, taken moments apart, for diastolic and SBP was used. The threshold for hypertension was set at 140/90 mmHg according to the NICE guidelines for hypertension management[45]. Both the diastolic and SBP reading above the threshold was required to be classified as hypertensive and increase the CVR score. Taking cholesterol or BP-lowering medication increased the CVR score. WHR was calculated as the ratio between waist circumference and hip circumference measured manually in centimetres. The threshold for WHR as a CVR was set according to WHO guidelines for each sex[46].

Diagnosed diabetes increased the CVR score by one. Smoking status included those who currently smoke or were past smokers based on pack year history, calculated as the daily number of cigarettes divided by 20 (pack size) and multiplied by the number of years smoking. The number of years smoking was estimated by the age an individual stopped smoking (or age at scan if current smoker) minus the age smoking was started. The number of pack years was adjusted for those who gave up smoking for more than 6 months. Important to our pack year calculation, the UK Biobank only collected information on number of cigarettes smoked in ex-smokers if they indicated that they smoked on "most or all days." An equivalent of 10–50 pack years increased the CVR score by 1 and greater than 50 pack years increased the score by 2. Carrying *APOE* ε3/3 alleles did not increase the CVR score, but ε4 carriage increased the score by one point for heterozygous carriers and two points for homozygous carries, based on the known risk to cardiovascular disease[47] and the development of sporadic dementia[48]. Genotyping was conducted by Affymetrix for UK Biobank using bespoke Axiom arrays[49].

**MRI data acquisition and analysis.** Imaging data were acquired on a Siemens Skyra 3T scanner with a 32-channel head coil. The full imaging protocol including acquisition details is openly available (https://biobank.ctsu.ox.ac.uk/crystal/docs/brain_mri.pdf). Of relevance to this study, a T1 weighted, 3D magnetisation-prepared rapid gradient echo sequence with 1 mm isotropic resolution, a T2 weighted fluid attenuated inversion recovery sequence ($1.05 \times 1 \times 1$ mm resolution) and a diffusion weighted, 2 mm isotropic voxel spin echo multiband echo-planar sequence, with 50 $b = 1000$ s mm and 50 $b = 2000$ s mm diffusion weighted volumes (100 encoding directions) were acquired. Imaging data were processed and quality checked using an automated pipeline that is openly available[50]. The pipeline produces imaging derived phenotypes (IDPs), summary information for each imaging modality. A subset of the IDPs provided by UK Biobank was used for the work in this paper. An overview of the analysis that had been used to generate the IDPs is given below.

**White matter hyperintensity load estimation**. The FLAIR image was gradient distortion and bias field corrected and linearly registered to the T1 image and to Montreal Neurological Institute (MNI 152) Atlas space. WMH segmentation was carried out using the Brain Intensity Abnormality Classification Algorithm (BIANCA)[51]. BIANCA is a fully automated method for classifying voxels based on relative intensity and spatial features (such as distance to ventricles). The output is a probability map that is thresholded at 0.8 to produce a binary map of lesion location and total WMH burden in mm³.

**Grey matter volume estimation**. The key stages of the T1 processing pipeline included correction for gradient distortion, registration to MNI 152 space, segmentation into grey matter, white matter and cerebrospinal fluid using FAST (fMRIB's automated segmentation tool) and bias field correction[50]. The volume of different tissue types and whole brain volume (grey and white matter) were estimated using SIENAX[52]. Regional cortical grey matter volumes were normalised for head size. We examined grey matter volume in frontoparietal cortex (Fig. 1c) using the canonical frontoparietal functional network derived by Yeo et al.[53]. The Yeo et al.[53], network parcellation is based on clustering performed on resting state fMRI data in 1000 subjects. We took the peak coordinates from each of the nodes in the frontoparietal control network (see Supplementary Table 3 for MNI coordinates and field IDs of IDPs) and found the IDPs that best spatially overlapped with this peak coordinate. For each region, grey matter volume was summed across hemispheres.

**Neurite orientation dispersion and density imaging**. Traditional measures of white matter microstructure such as fractional anisotropy and mean diffusivity are based on a crude diffusion tensor model that cannot distinguish contributions to white matter structure from intracellular and extracellular water. These indices also lack sensitivity to complex crossing fibre architecture present in over 90% of white matter connections[54]. Multicompartment models such as neurite orientation dispersion and density imaging (NODDI) aim to be more biologically plausible than simple ellipsoid-based tensor models and estimate neurite density (intracellular volume fraction (ICVF)) and measures of tract coherence and complexity (orientation dispersion (OD))[55].

Diffusion weighted images were corrected for eddy currents and head motion using FSL's eddy tool and then gradient distortion corrected. NODDI modelling was implemented using the AMICO (accelerated microstructure imaging via convex optimisation) toolbox to generate IDPs representing ICVF and OD. NODDI maps were aligned to a standard space white matter skeleton using optimised registration methods[50]. IDPs for 48 tracts derived from the John Hopkins University Tractography Atlas were generated by averaging NODDI indices within each mask. The IDPs representing NODDI indices for the superior longitudinal fasciculus were extracted for use here (Fig. 1d). The superior longitudinal fasciculus is a major associative tract known to support the frontoparietal network in monkeys and humans[30,35]. ICVF is the fraction of restricted vs. unrestricted diffusion and a measure of neurite density. OD is a measure of angular variation indexing the configuration of axons in which smaller values are associated with tighter bundles (Fig. 1b). Head size and scan-date-derived confounds were regressed out of all imaging IDPs before further analyses as they are known confounds[56].

**Statistical analysis**. Calculations were performed in Matlab R2018a or in R, using the Lavaan package[57] for SEM. For preprocessing, all variables were median absolute deviation normalised. Total WMH load was cube root transformed to correct for a heavily skewed distribution. Three latent variables were estimated using CFA. A major advantage of the use of latent variables is control of measurement error, which can artificially reduce the relationship between measured variables in standard univariate analyses but this effect is strongly attenuated in SEM[58]. Latent variables also allow a continuous representation of multiple measures, which may not be easily combined otherwise. A latent variable representing speed of processing and executive function was estimated (performance in the simple reaction time and pairs matching task). Speed of processing and executive function are the cognitive domains that show the steepest declines in aging[6] and are most frequently associated with cerebrovascular burden[36]. Given their frequent covariance we reasoned that they may be indexing a latent variable, which for simplicity we termed, executive function. Latent variables for white matter integrity of the superior longitudinal fasciculus (OD and ICVF), and grey matter integrity (IDPs representing grey matter volume in the frontoparietal network) were estimated in the same model (Supplementary Fig. 1). There are reliable and robust decreases in fractional anisotropy with increasing age, but the measure is not specific to the underlying cause of integrity changes. The NODDI measures of ICVF and OD index neurite density and tract complexity, respectively. A recent, in depth analysis of the microstructural measures showed that 75.71% of the relationship between age and FA was mediated by ICVF and OD[59]. Our latent variable for white matter integrity therefore included ICVF and OD as sensitive measures of white matter integrity within the superior longitudinal fasciculus. Finally, our latent variable representing grey matter volume of the frontoparietal network was based on the principle of network level structural covariance, which assumes covariance amongst regions within the same network.

Multiple regression was used to examine the relationship between individual CVR factors and the Executive Function latent variable, adjusted for age and socio-economic status, as estimated by the Townsend Deprivation Index. The Townsend Deprivation Index is a measure of deprivation based on unemployment, home and car ownership and household overcrowding calculated according to participant postcode. The regression model was tested for multicollinearity using variance inflation factor. We set the threshold for statistical significance at $p < 0.001$.

The relationship between SBP and executive function was tested between individuals on antihypertensive medications and those not on antihypertensive medications. We used, being on hypertension medication, as a proxy for established hypertension. While similar numbers of individuals provided a self-report of hypertension, use of antihypertensive medication provided a hard index of hypertension that was less likely to be affected by recall bias and would be supported by criteria recommended by current UK NICE guidance using either ambulatory BP monitoring, home automated BP monitoring or, previously, repeated measures over several assessments following a single high reading before treatment initiation[60].

We used a model-free sliding window approach to study the relationships between the executive function latent variable and SBP, and with age. This method does not assume a linear relationship. A systolic window of observations (each containing 10% of the participants) of fixed systolic-quantile width was moved along the SBP distribution (code: conditionalPlot.m available here: https://osf.io/vmabg/)[61,62]. We applied a smoothing Gaussian kernel of 10 using the moving average method. The results from using ten fixed systolic bins, with 16 mmHg widths (rather than fixed number of observations), are shown as Supplementary Fig. 2. Pearson correlations were calculated between executive function and SBP, and Bonferroni-correction was applied. Comparison of correlation coefficients between participants on antihypertensives and those not on antihypertensives were done following Fisher's $r$ to $z$ transformation. Deconfounding age was done using age residuals within the quantile bins described above.

Path modelling was used to estimate the directional dependencies between the latent variables and observed variables. Observed variables were WMH load (total lesion volume), age (at MRI scan) and total CVR score. Correlated residual paths identified from modification indices were included in the model. We included correlated error between volumes in the grey matter volume latent variable on the basis that some of these regions may have participation in other structural covariance networks beyond the frontoparietal network. Missing data were assumed to be missing at random and was estimated using full information maximum likelihood, which gives unbiased parameter estimates and standard errors. We compared our hypothesised structural and path model to a model excluding grey matter volume and a model excluding white matter integrity (by fixing all paths that included these variables to zero). An identical SEM replacing the frontoparietal network regions and SLF with regions and tracts associated with the language network was used to test the specificity of our findings to the frontoparietal network (Supplementary Fig. 3, Table 5).

Semi-partial correlations and mediation analyses were conducted to estimate the directional influence of brain imaging variables to executive function for the purposes of clarifying the mechanisms underlying executive dysfunction in healthy aging. Mediation analysis was run in Lavaan, with nonparametric bootstrapping with 10,000 iterations to estimate direct and indirect effects between variables.

**Reporting summary**. Further information on research design is available in the Nature Research Reporting Summary linked to this article.

## Data availability

The data analysed during the current study are available from the UK Biobank https://www.ukbiobank.ac.uk/researchers/. The variables used are detailed in Supplementary Table 2. Code for the sliding window analyses is available from https://osf.io/vmabg/.

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

## Acknowledgements

This work was funded by the Wellcome Trust and MRC (Wellcome Trust Principal Research Fellowship to M.H. and MRC Clinician Scientist Fellowship to S.M.), and by the Oxford NIHR Biomedical Research Centre.

## Author contributions

Design, analysis, paper writing, critical revisions (M.V. and X.Y.T.); preprocessing and analysis of data (J.P.); design, statistical analysis and critical revisions of the paper (T.N., S.S., S.M., M.H.)

## Competing interests

The authors declare no competing interests.

**Additional information**

