## [Peer Review File · Nature Communications]

REVIEWER COMMENTS

Reviewer #1 (Remarks to the Author):

The manuscript responds to an important biological and clinical question (cerebrovascular risk factors impact on cognitive deterioration). The sample size is impressive, while all the statistical analyses are properly formulated and performed, providing robust interesting conclusions, which are extensively discussed by the authors in biological and clinical terms.

I don't have any major or minor comments that could substantially improve the manuscript. Thus, I suggest direct acceptance of the manuscript for its publication.

Reviewer #2 (Remarks to the Author):

This is a well written report on the link between cardiovascular risk factors and cognitive function and brain MRI findings.

The authors employ the UK Biobank and study cardiovascular risk factors (in isolation and conjunction) in relation to cognition and then employ structural equation models to study pathway effects of brain MRI towards cognitive function.

My main point of critique relates to the novelty of the findings.

The link between cardiovascular factors and brain health has been established for over 2 decades now and many study have also shown mid-life effects, treatment effects and mediating effects (of brain structure). The large sample size of the UKBB is obviously a strength but it is unclear how large sample size (that merely relates to increasing power) could unravel novel findings not already known. The authors should therefore more explicitly state what knowledge gap is being addressed and what insight is provided that could not already be inferred from available literature.

Another point relates to the conclusion on hypertension treatment. The authors should tone down their interpretation on treatment windows, since this is a cross-sectional analysis and the step towards treatments is too far to be made with the current findings.

We would like to extend our thanks to the reviewers for their time assessing our manuscript. We have addressed the points raised by Reviewer #2, point by point below. Major changes to sections of the text are shown in red type.

Reviewer #1 (Remarks to the Author):

The manuscript responds to an important biological and clinical question (cerebrovascular risk factors impact on cognitive deterioration). The sample size is impressive, while all the statistical analyses are properly formulated and performed, providing robust interesting conclusions, which are extensively discussed by the authors in biological and clinical terms.

I don't have any major or minor comments that could substantially improve the manuscript. Thus, I suggest direct acceptance of the manuscript for its publication.

We thank reviewer 1 for their very encouraging comments and recognition of the importance of our work.

Reviewer #2 (Remarks to the Author):

This is a well written report on the link between cardiovascular risk factors and cognitive function and brain MRI findings.

The authors employ the UK Biobank and study cardiovascular risk factors (in isolation and conjunction) in relation to cognition and then employ structural equation models to study pathway effects of brain MRI towards cognitive function.

My main point of critique relates to the novelty of the findings.

The link between cardiovascular factors and brain health has been established for over 2 decades now and many study have also shown mid-life effects, treatment effects and mediating effects (of brain structure). The large sample size of the UKBB is obviously a strength but it is unclear how large sample size (that merely relates to increasing power) could unravel novel findings not already known. The authors should therefore more explicitly state what knowledge gap is being addressed and what insight is provided that could not already be inferred from available literature.

We thank the reviewer for their insightful comments and the opportunity to make the novel contributions of our work clearer.

While it is true that the relationship between cardiovascular risk factors and brain health have been established over two decades, our study makes several advances on this literature as outlined below:

1. Much of the existing literature focuses on cardiovascular risk factors associated with the long-term risks of *developing dementia*. The impact of these risk factors, or their modification, on *current or shorter-term* cognitive performance across the lifespan is far less extensively investigated. Where it has been examined, the studies focus on identifying a significant difference in global cognition – by comparing group mean total scores on clinical tests such as the mini-mental state examination (MMSE) – in relation to the absence or presence of a cardiovascular risk factor such as diabetes or hypertension. Meta-analyses have

actually shown no effect or inconclusive evidence of the effect of hypertensive treatment on cognitive performance in this context. However, only global cognition was measured using the MMSE which may obscure effects on executive function which is known to be impacted by cardiovascular risk.

Here we use a more sensitive parametric analysis to show a continuous relationship between increasing systolic blood pressure and poorer *current* executive function.

The revised Introduction now includes the following:

In addition to these disparate findings relating to long-term dementia risk, the relationship between cerebrovascular risk factors and current or shorter-term cognitive function in people without dementia or MCI also remains unclear. Several studies have suggested an association between cerebrovascular burden and cognition in healthy cohorts. However, most investigations have assessed the presence of vascular risk factors or risk factor modification and related these to global cognition, using either MMSE or composite cognitive scores^{13–16}. One recent study has reported a benefit of statin and blood pressure-lowering medication using a more sensitive digit-symbol substitution score in healthy people without any known cardiovascular disease¹⁷. Overall, however, to the best of our knowledge there has been no clear demonstration of a parametric effect of risk factors (e.g., BP level, rather than simply presence or absence of hypertension or BP lowering treatment) associated with systematic differences in cognitive performance. This represents an important gap in current knowledge.

In addition, the strength of any conclusion regarding the impact of cerebrovascular risk factors – or their modification – on cognition would be most convincing if they were shown to be accompanied by changes in brain structure, ideally in a large sample in which cognitive function was concurrently measured with tests that are more sensitive than the MMSE or other global measures of cognition. However, to date, we lack data from a large-scale investigation which combines sensitive measure of cognition with modern brain imaging markers to provide more definitive conclusions.

2. The demonstration of the parametric effect of systolic blood pressure on a continuous score of executive function was made possible by two unique factors of this study: the large sample size of the UK Biobank and the use of a latent variable based on neuropsychological testing which is likely more sensitive than traditionally used total cognitive test scores.

The revised discussion now includes the following:

Individuals who were on antihypertensives, i.e., with an established diagnosis of hypertension, showed an ‘effective systolic range’ below ~140 mmHg corresponding to stable executive function while, above this threshold, increasing SBP was associated with a decline in executive function. **Demonstrating this threshold and effect of SBP, in a parametric manner, on a continuous score of executive function was made possible by two unique factors of this study: the large sample size of the UK Biobank and the use of a latent variable based on neuropsychological testing which is likely more sensitive than traditionally used total cognitive test scores.**

3. In contrast to having group boundaries for blood pressure control, our findings show a potential target for blood pressure management (< 140 mmHg) to optimise cognitive health. Above this level, executive function deteriorated, while below it there was no additional benefit associated with lower blood pressure. This has potentially important clinical implications for aggressive risk factor management as well as avoiding over-treatment.

The revised Discussion now includes:

The findings of the current study have the potential to provide meaningful evidence on BP targets for cognitive health in normal ageing. They are in accord with recent findings which emphasize the importance not simply to treat, but to treat effectively and monitor for response. The prospective SPRINT-MIND study³⁷ released results on the impact of intensive vs standard BP control (defined as <120 mmHg vs. 120-140 mmHg respectively) on cognitive impairment risk ($N= 9361$). This trial was terminated early and was considered underpowered. However, available results suggested no significant reduction in incidence of MCI³⁷ from aggressive BP control. **The results presented in our investigation would be consistent. They show, in a parametric manner, that controlling SBP below 140 mmHg had no added benefit on executive function. Unlike previous studies which use BP levels to categorize groups, the findings presented here, using BP as a continuous variable, offer insights on a precise blood pressure target for optimal cognitive health, albeit with the caveats associated with a cross-sectional study. They have potentially important implications for rational treatment guidance. SBP levels above 140 mmHg are associated with a measurable detrimental impact on current executive function. Conversely, systolic levels below 140 mmHg confer no additional benefit. Therefore, very aggressive treatment of BP risks hypotensive and other side-effects, without necessarily providing a positive impact on cognition.**

4. Although mid-life effects of cardiovascular risk factors have been shown, our results show *both* mid-life effects, in a parametric manner, *as well as* the effects of hypertensive treatment that reveal a systolic BP range for optimal cognitive performance at a population level. This effect was not present in late-life despite anti-hypertensive treatment scores.

The revised Discussion now states:

Further age-group analysis examining mid- and late-life groups showed striking differences. SBP predicted executive function within the *mid-life* group, but not in late-life individuals (aged >70 years). **While previous studies have suggested that higher systolic levels in mid-life may lead to worse cognitive performance in late-life^{15,38}, our results show the parametric benefit of hypertension treatment on current cognitive health in mid-life, but no significant effect of antihypertensives in late-life. This finding, in combination with the results of a recent study that examined dementia incidence associated with mid- to late-life BP patterns⁸, emphasizes the potential importance of**

treating mid-life hypertension for current cognitive function as well as for later dementia risk.

5. As well as providing more power, our large sample size allows us to perform complex multivariate modelling. Existing studies have examined the relationship between cardiovascular risk and brain health, or between brain health and cognition or cognition and individual cardiovascular risk factors. Given our large sample, we are able to put all of these variables into one large, complex model that allows us to examine multiple simultaneous relationships and establish how cerebrovascular risk and grey *and* white matter network integrity relates to cognitive decline in healthy ageing. Furthermore, although previous studies have examined general brain health and global cognition, we examine the impact of cerebrovascular risk factors on domain specific cognitive function (executive function) as it relates to a specific brain network.

The revised Introduction now states:

Previous studies have examined the relationship between either cardiovascular risk and brain health, or between brain health and cognition, or between cognition and individual cardiovascular risk factors. Given our large sample, we were able to include all these variables into one large, complex model that allowed us simultaneously to examine multiple relationships and establish how cerebrovascular risk and grey *and* white matter network integrity in a frontoparietal network relates to cognitive decline, specifically with respect executive function, in healthy ageing.

6. The reviewer's suggestion to make more explicit the 'knowledge gap' is extremely helpful. The Introduction now includes the following:

To summarise, the existing literature leaves several important unanswered questions. What are the effects of cerebrovascular risk factors on cognitive function in healthy people across the lifespan? And is it possible to show a relationship, in a parametric manner, between a risk factor such as blood pressure level and cognition? Does treatment of a modifiable risk factor, such as hypertension, affect cognitive function? If so, is there a range of blood pressure over which this treatment has an impact? Finally, does the integrity of a distributed brain network underlying a specific cognitive function explain decline in that function across the lifespan as a result of cerebrovascular burden?

Another point relates to the conclusion on hypertension treatment. The authors should tone down their interpretation on treatment windows, since this is a cross-sectional analysis and the step towards treatments is too far to be made with the current findings.

We agree with the reviewer that our cross-sectional analyses require confirmation from longitudinal analyses (which will be available, and which we intend to analyse in future Biobank releases) before we can make treatment recommendations.

We have offered relevant longitudinal evidence from other studies (predominantly addressing dementia risk) which support our findings and suggest robust and precise risk factor management.

The revised Discussion includes:

While previous studies have suggested that higher systolic levels in mid-life may lead to worse cognitive performance in late-life^{15,38}, our results show the parametric benefit of hypertension treatment on current cognitive health in mid-life, but no significant effect of antihypertensives in late-life. This finding, in combination with the results of a recent study that examined dementia incidence associated with mid- to late-life BP patterns⁸, emphasizes the potential importance of treating mid-life hypertension for current cognitive function as well as for later dementia risk.

We have also toned down our interpretation, as suggested by the reviewer. For example, we have now removed the final sentence “which should inform public health clinical guidance and practice” and instead concluded with the following:

Future release of prospective UK Biobank data will add a valuable temporal dimension to our interpretation, including longitudinal effect of antihypertensives on executive function and additional dementia outcomes.

In the rapidly growing social and health context of our ageing society, the findings presented here demonstrate the relationship between key cerebrovascular risk factors and healthy cognitive ageing, within the UK Biobank cohort, with implications on lifestyle practices for middle- and late-age populations. **Moreover, the results are suggestive of targeted and time-sensitive blood pressure control being beneficial to cognitive and brain health.**

We thank the reviewer for their valuable suggestions which have helped to improve our manuscript.

REVIEWERS' COMMENTS:

Reviewer #1 (Remarks to the Author):

As in my first review of this manuscript, and based on its high quality and the analyzed topic's relevance, I suggest its publication. I didn't have major critics or concerns with the first version, neither with this improved version. It is, in my opinion, an excellent and complete manuscript.

Reviewer #2 (Remarks to the Author):

I appreciate the efforts by the authors to demonstrate the knowledge gap(s) addressed by the current study.

However, I am not convinced by several of their claims, since those could be easily falsified. A quick and rough query in PubMed already reveals many studies that have investigated continuous measures of vascular risk factors in relation to various cognitive domains (besides MMSE) and to brain MRI measures (in the same populations, but reported in different publications).

For instance:

Euser SM et al. JAGS 2009

Birns and Kalra. J Hum Hypertension 2008 (review)

Liu A et al. Alz Dis Assoc Dis 2019

Van Dijk E et al. Hypertension 2004

A more systematic search would reveal studies that would allow synthesis of the same conclusions as in the current manuscript.

With all this background information, I still am doubtful what the added value is of this study beyond the huge body of literature already available (apart from the sample size)

We would like to extend our thanks to the reviewers for their time assessing our manuscript.

We note the general remark of Reviewer #2 which suggests that the only novelty of our manuscript is the very large sample size. While the latter is true and gives confidence in our findings, it is incorrect to conclude that this is the only novel aspect of our study. We explain in detail below why the papers cited by the Reviewer did not in fact perform similar analyses to our own. In brief, none of those previous studies report the following combination of findings:

- Impact of blood pressure of cardiovascular risk factors on cognition *in middle age*
- Use *continuous measures* of blood pressure (BP)
- Employ *continuous measures* of cognitive or executive function
- Relate these to *brain imaging* in middle age using objective measures of white matter integrity

It is true that in some form or other, they did examine one of the features noted above, but they certainly did not bring these altogether to give confidence that they are related in the way we demonstrate here.

Reviewer #2 (Remarks to the Author):

I appreciate the efforts by the authors to demonstrate the knowledge gap(s) addressed by the current study.

However, I am not convinced by several of their claims, since those could be easily falsified. A quick and rough query in PubMed already reveals many studies that have investigated continuous measures of vascular risk factors in relation to various cognitive domains (besides MMSE) and to brain MRI measures (in the same populations, but reported in different publications).

For instance:

Euser SM et al. JAGS 2009

Birns and Kalra. J Hum Hypertension 2008 (review)

Liu A et al. Alz Dis Assoc Dis 2019

Van Dijk E et al. Hypertension 2004

A more systematic search would reveal studies that would allow synthesis of the same conclusions as in the current manuscript.

With all this background information, I still am doubtful what the added value is of this study beyond the huge body of literature already available (apart from the sample size)

We would like to take the opportunity to clarify our points and examine the references provided.

Euser et al. (2009) examined the Rotterdam and Leiden 85- plus cohort. They showed that increasing systolic and diastolic blood pressure (by bins of 10 mmHg) at time of initial measurement was related to worse cognitive function using psychometric testing **performed 11 years later**. The longitudinal nature of this study allowed valuable insight at cognitive status at a later stage. However, this study **did not examine the association of blood pressure at the time of measurement to the mmHg resolution that was done using the UK Biobank cohort. Although it adjusted for the**

impact of cardiovascular risk factors their impact was not examined. Nor did this study include neuroimaging data.

Birns & Kalra (2009) performed an excellent review of the literature regarding cognition and blood pressure to that point. A wide range of cross-sectional studies had examined cognitive ability using the Mini Mental State Examination (MMSE) and psychometric testing. However, other than lower sample size, most studies focussed on the impact of high and normal blood pressure using arbitrarily set group boundaries (based on guideline parameters) – **not continuous measures of BP. All but five studies examined older cohorts of above 65 years of age. The assessment of brain structural or functional analysis was not included in this review.**

Liu et al. (2019) describes an interesting approach to examining non-linear associations between memory and blood pressure. The authors examine a smaller number of individuals above age 65 years divided into three groups of being hypertensive, normotensive and hypotensive. They show that, among normotensive individuals, those with blood pressure around 140/80 was associated with the highest memory scores while, among hypertensive individuals, higher blood pressure around 150/80 predicted lower memory scores. This was shown using non-linear data analysis but was not evident from examining linear associations. **Executive function was not examined specifically and there was no brain imaging analysis included in this study.**

Van Dijk et al. (2008) examined the relationship between subcortical white matter lesions on MR imaging with blood pressure measurements in ten cohorts with individuals between 65 and 75 years. Single blood pressure measurements and increase in blood pressure over time was associated with increasing white matter lesion burden **based on a subjective visual rating, not objective analysis methods as we used.** The authors used a logistic regression analysis to show that individuals with poorly controlled hypertension had a higher risk of severe white matter lesions. This was an important study examining the effect of blood pressure on developing white matter lesions in the brain **but did not examine the effect on cognition.**

While we agree with the reviewer that there is a large body of literature available which reflects the importance of this area, our study, to the very best of our knowledge, uses a combination of methods which previously has not been deployed in a large sample size, including people in middle age. The results are the strongest evidence to date that control of vascular risk factors in middle age might be important for cognitive function **at that age, not simply later in life.**

The revised manuscript now includes all the references mentioned above by the Reviewer, but points out that no previous study has performed the combination of analyses we report:

Notwithstanding these limitations, the results from several different cohorts suggest that cerebrovascular risk factors might be an important target to protect brain health and thereby reduce

the risk of developing dementia. For example, a community-based study followed 6330 non-demented individuals, aged 45-99 years, and showed that midlife diabetes and impaired glucose tolerance conferred increased risk of dementia^{7,8}.

In addition to these disparate findings relating to long-term dementia risk, the relationship between cerebrovascular risk factors and current cognitive function in people without dementia or MCI also remains unclear. Several studies have suggested an association between cerebrovascular burden and cognition in healthy cohorts (reviewed, for example here¹⁴). However, most investigations have assessed the presence of vascular risk factors or risk factor modification and related these to global cognition, using either MMSE or composite cognitive scores¹⁵⁻¹⁹. One recent study has reported a benefit of statin and blood pressure-lowering medication using a more sensitive digit-symbol substitution score in healthy people without any known cardiovascular disease¹⁷. Overall, however, to the best of our knowledge there has been no clear demonstration of a parametric effect of risk factors (e.g., BP level, rather than simply presence or absence of hypertension or BP lowering treatment) associated with systematic differences in cognitive performance. This represents an important gap in current knowledge.

It is, of course, well established that cerebrovascular risk factors are associated closely with the macroscale marker white matter hyperintensities (WMH)²¹ on magnetic resonance imaging (MRI). WMH have generated significant interest as a key marker of cerebrovascular burden in the ageing brain^{22,23}.

While we appreciate the overall general point of the Reviewer, we would be grateful for an executive editorial decision on this manuscript, given the long delays already incurred. We are also keen to post a press release on the findings soon because the results are of fundamental, general interest.